Cyclic mechanical stretch down-regulates cathelicidin antimicrobial peptide expression and activates a pro-inflammatory response in human bronchial epithelial cells

Karadottir Harpa 1
Kulkarni Nikhil Nitin 1
Gudjonsson Thorarinn 2 3
Karason Sigurbergur 4
Gudmundsson Gudmundur Hrafn 1 ghrafn@hi.is
1 Biomedical Center and Department of Life and Environmental Sciences, University of Iceland , Reykjavik , Iceland
2 Stem Cell Research Unit, Biomedical Center, Department of Anatomy, Faculty of Medicine, School of Health Sciences, University of Iceland , Reykjavik , Iceland
3 Department of Laboratory Hematology, Landspitali University Hospital, University of Iceland , Reykjavik , Iceland
4 Department of Anaesthesia and Intensive Care and Faculty of Medicine, Landspitali University Hospital and University of Iceland , Reykjavik , Iceland
Goyal Pankaj
Electronic publication date: 2015 Dec 7
Publication date: 2015
Volume: 3
Electronic Location ID: e1483
Received 2015 Sep 21; Accepted 2015 Nov 18
Copyright: © 2015 Karadottir et al.
Copyright year: 2015
Copyright holder: Karadottir et al.
License: This is an open access article distributed under the terms of the Creative Commons Attribution License, which permits unrestricted use, distribution, reproduction and adaptation in any medium and for any purpose provided that it is properly attributed. For attribution, the original author(s), title, publication source (PeerJ) and either DOI or URL of the article must be cited.
License URL: https://creativecommons.org/licenses/by/4.0/

Keywords: LL-37, Vitamin D3, Pro-inflammatory response, Innate immunity, Ventilator induced lung injury

Funding: Landspitali University Hospital Össur hf and the Oddur Olafsson fund University of Iceland Icelandic Research Fund (RANNIS) This project was supported by grants from LSH (Landspitali University Hospital), Össur hf and the Oddur Olafsson fund to Sigurbergur Karason. In addition this work was supported by funds from the University of Iceland and RANNIS. Nikhil Nitin Kulkarni was supported by University of Iceland grant for PhD students and RANNIS. The funders had no role in study design, data collection and analysis, decision to publish, or preparation of the manuscript.

==============================
Mechanical ventilation (MV) of patients can cause damage to bronchoalveolar epithelium, leading to a sterile inflammatory response, infection and in severe cases sepsis. Limited knowledge is available on the effects of MV on the innate immune defense system in the human lung. In this study, we demonstrate that cyclic stretch of the human bronchial epithelial cell lines VA10 and BCi NS 1.1 leads to down-regulation of cathelicidin antimicrobial peptide (CAMP) gene expression. We show that treatment of VA10 cells with vitamin D3 and/or 4-phenyl butyric acid counteracted cyclic stretch mediated down-regulation of CAMP mRNA and protein expression (LL-37). Further, we observed an increase in pro-inflammatory responses in the VA10 cell line subjected to cyclic stretch. The mRNA expression of the genes encoding pro-inflammatory cytokines IL-8 and IL-1β was increased after cyclic stretching, where as a decrease in gene expression of chemokines IP-10 and RANTES was observed. Cyclic stretch enhanced oxidative stress in the VA10 cells. The mRNA expression of toll-like receptor (TLR) 3, TLR5 and TLR8 was reduced, while the gene expression of TLR2 was increased in VA10 cells after cyclic stretch. In conclusion, our in vitro results indicate that cyclic stretch may differentially modulate innate immunity by down-regulation of antimicrobial peptide expression and increase in pro-inflammatory responses.

Introduction

Mechanical ventilation (MV) is a lifesaving treatment for patients suffering from severe respiratory failure by alleviating the work of breathing and facilitating alveolar gas exchange (Slutsky & Ranieri, 2013). MV has, however, been associated with side effects including ventilator induced lung injury (VILI) coupled with injury on lung tissue, stress on epithelial and endothelial barriers, apoptosis, pro-inflammatory responses, increased oxidative stress and secondary infections like nosocomial bacterial pneumonia. This can be followed by sepsis or systemic inflammatory response syndrome and increased mortality (Baudouin, 2001; Uhlig, 2002; Syrkina et al., 2008). Success of treatment with MV requires limitation of VILI and associated side effects (Fan, Villar & Slutsky, 2013). This can be accomplished by either decreasing mechanical stress produced by MV or by increasing the endurance of lung tissues to such strain. Hence, it has become imperative to study the molecular mechanisms behind VILI in details to improve outcomes in patients treated with MV. Although poorly defined, down-regulation of innate immune responses has been proposed to favor bacterial growth and development of ventilator associated pneumonia (VAP) in the lungs of patients during MV (Santos et al., 2005).

Antimicrobial polypeptides (AMPs) constitute an important arm of the innate immune defense in the lungs and are expressed ubiquitously in epithelial cells, neutrophils and monocytes or macrophages (Laube et al., 2006). These cationic polypeptides are categorized into: (1) smaller processed peptides such as cathelicidins and defensins and (2) larger polypeptides like lactoferrin, lysozyme and secretory leukocyte peptidase inhibitor (SLPI) (Laube et al., 2006). LL-37 is the main cathelicidin antimicrobial peptide (CAMP) in humans, encoded by the CAMP gene (Dürr, Sudheendra & Ramamoorthy, 2006). LL-37 is stored as a pro-form (pro-LL-37) in cells and is activated upon secretion to the mature form LL-37 by specific proteases (Sørensen et al., 2001). LL-37 has direct antimicrobial activity against multiple pathogens and has been demonstrated to exhibit pro- and anti-inflammatory responses, wound healing and angiogenic properties (Cederlund, Gudmundsson & Agerberth, 2011). Inducers of AMPs like vitamin D3 (1, 25-dihydroxy vitamin D3 or 1,25D3) and 4-phenyl butyric acid (PBA) have been shown to increase CAMP gene expression via the vitamin D receptor (VDR) (Gombart, Borregaard & Koeffler, 2005; Kulkarni et al., 2015a; Kulkarni et al., 2015b). A recent clinical trial demonstrated that lower vitamin D3 levels and cathelicidin expression was associated with higher mortality in critically ill patients usually receiving MV (Leaf et al., 2015). The effects of MV on respiratory cells can be modeled in vitro by applying defined cyclic mechanical stretch mimicking the frequency and stretch conditions during MV (Pugin et al., 2008; Wu et al., 2013).

In this study, we demonstrate that cyclic mechanical stretch of human bronchial epithelial cells VA10 and BCi down-regulates the expression of antimicrobial peptide cathelicidin. Treatment with AMP inducers vitamin D3 and/or PBA counteracted cyclic stretch mediated down-regulation of cathelicidin expression in VA10 cells. We further demonstrate that cyclic stretching of VA10 cells activated a pro-inflammatory response by enhancing expression of pro-inflammatory cytokines and increasing oxidative stress.

Materials and Methods

Cell culture, reagents and cyclic stretch

An E6/E7 viral oncogene immortalized human bronchial epithelial cell line VA10 was cultured as described previously (Halldorsson et al., 2007). Briefly, the cells were maintained in Bronchial/Tracheal Epithelial cell growth medium (Cell Applications, San Diego, CA, USA) with Penicillin-Streptomycin ((20 U/ml, 20 µg/ml, respectively) (Life Technologies, Carlsbad, CA, USA)) at 37 °C and 5% CO2. BCi. NS 1.1 (henceforth referred to as BCi) is a human bronchial epithelial cell line was a kind gift from Dr. Matthew S. Walters, Weill Cornell Medical College, New York NY, USA (Walters et al., 2013) and was established by immortalization with retrovirus expressing human telomerase (hTERT). The BCi cells were cultured as described above for VA10 cell line. Equal amount of cells were seeded on each well in a 6 well collagen I coated Bioflex plates (Flexcell International Corporation, Burlington, CA, USA), and grown to approximately 80% confluence. These plates were then transferred to a base plate of the cell stretching equipment Flexcell FX-5000TM Tension System (Flexcell International Corporation, Burlington, CA, USA) in a humidified incubator at 37 °C and 5% CO2. The cells were subjected to cyclic mechanical stretch with the following parameters: a stretching rate of 20% with a square signal, 0.33 Hz frequency (20 cycles/min) and a 1:1 stretch:relaxation ratio, as described previously (Pugin et al., 2008). The cells were stretched for 6 h and 24 h as described in the results. Control Bioflex plates were kept in the same incubator under static conditions as non-stretch controls. Vitamin D3 (1,25D3) and Sodium 4-phenyl butyric acid (PBA) were purchased from Tocris bioscience, UK. Vitamin D3 was reconstituted in 100% ethanol as per manufacturer’s instructions. The final concentration of the solvent was kept at 0.2% v/v and did not affect gene and protein expression of target genes. PBA was reconstituted in ultrapure H2O.

RNA isolation and quantitative real time PCR

Total RNA was isolated with NucleoSpin RNA kit (Macherey-Nagel, Düren, Germany) and quantified on a spectrophotometer (Nanodrop, Thermo Scientific, Waltham, MA, USA). One µg of total RNA was reverse transcribed into first strand cDNA for each sample with a RevertAid First strand cDNA synthesis kit (Thermo Scientific, Waltham, MA, USA) and modified with 100 unit of reverse transcriptase per reaction. Power SYBR® green Universal PCR master mix (Life technologies, Carlsbad, CA, USA) was used to quantify the cDNA on a 7500 Real time PCR machine (Life technologies, Carlsbad, CA, USA). The reference gene used for all experiments was UBC (Ubiquitin C) and PPIA (Peptidylprolyl Isomerase A) and an arithmetic mean of reference gene Ct values was used. Primers for TLR1 and TLR6 were designed with Pearl primer and used at a final concentration of 300 nM (Marshall, 2004). All other primers were purchased from Integrated DNA technologies (PrimeTime Predesigned qPCR Assay) and used at a final concentration of 500 nM as per manufacturer’s instructions. The qPCR cycling conditions were as follows; (1) Hold stage: 95 °C for 10 min, followed by 40 cycles of (2) De-natured stage: 95 °C for 15 s and (3) Annealed/extended stage: 60 °C for 1 min. The 2(−ΔΔCT) Livak method was utilized for calculating fold differences over untreated control (Livak & Schmittgen, 2001). A detailed list of the primers used in the q-RT-PCR assay is shown in Table 1.

Table 1 Primers used in the q-RT PCR assay.

Primer	Gene symbol	Ref. seq. number	Forward primer	Reverse primer	
CAMP	CAMP	NM_004345	5′-GCA GTC ACC AGA GGA TTG TGA C-3′	5′-CAC CGC TTC ACC AGC CC-3′	
DEFB1	DEFB1	NM_005218	5′-CCA GTC GCC ATG AGA ACT CC-3′	5′-GTG AGA AAG TTA CCA CCT GAG GC-3′	
IL-8	CXCL8	NM_000584	5′-CTG GCA TCT TCA CTG ATT CTT G-3′	5′-TGT CTG GAC CCC AAG GAA-3′	
IP-10	CXCL10	NM_001565	5′-CAG TTC TAG AGA GAG GTA CTC CT-3′	5′-GAC ATA TTC TGA GCC TAC AGC A-3′	
RANTES	CCL5	NM_002985	5′-TGC CAC TGG TGT AGA AAT ACT C-3′	5′-GCT GTC ATC CTC ATT GCT ACT-3′	
IL-1β	IL1B	NM_000576	5′-GAA CAA GTC ATC CTC ATT GCC-3′	5′-CAG CCA ATC TTC ATT GCT CAA G-3′	
PPIA	PPIA	NM_021130	5′-TCT TTC ACT TTG CCA AAC ACC-3′	5′-CAT CCT AAA GCA TAC GGG TCC-3′	
TLR1	TLR1	NM_003263	5′-CAA GAC TGT AGC AAA TCT-3′	5′-GTT TCG CCA GAA TAC TTA-3′	
TLR2	TLR2	NM_003264	5′-ATG ACC CCC AAG ACC CA-3′	5′-CCA TTG CTC TTT CAC TGC TTT C-3′	
TLR3	TLR3	NM_003265	5′-GCA CTG TCT TTG CAA GAT GA-3′	5′-AGA CCC ATA CCA ACA TCC CT -3′	
TLR4	TLR4	NM_003266	5′-ACC CCA TTA ATT CCA GAC ACA-3′	5′-GAG TAT ACA TTG CTG TTT CCT GTT G-3′	
TLR6	TLR6	NM_006068	5′-TAT CCT ATC CTA TTG-3′	5′-AGT TGC CAA ATT CCT TAC-3′	
TLR8	TLR8	NM_138636	5′-GAT CCA GCA CCT TCA GAT GAG-3′	5′-ACT TGA CCC AAC TTC GAT ACC-3′	
TLR9	TLR9	NM_017442	5′-GGA GCT CAC AGG GTA GGA A-3′	5′-AGA CCC TCT GGA GAA GCC-3′	
UBC	UBC	NM_021009	5′-GAT TTG GGT CGC AGT TCT TG-3′	5′-CCT TAT CTT GGA TCT TTG CCT TG-3′	
LZY	LZY	NM_000239	5′-CTC CAC AAC CTT GAA CAT ACT GA-3′	5′-AGA TAA CAT CGC TGA TGC TGT AG-3′	
LTF	LTF	NM_001199149	5′-AAT AGT GAG TTC GTG GCT GTC-3′	5′-TGT ATC CAG GCC ATT GCG-3′	

Sodium dodecyl sulfate-polyacrylamide gel electrophoresis (SDS-PAGE) and western blot analysis

Supernatants from treated cells were enriched for proteins on Oasis HLB mini columns (Waters, Milford, MA, USA) and lyophilized (Speed Vac; Thermo Scientific, Waltham, MA, USA). Ten µg of total lyophilized protein was loaded on a 4–12% Bis–Tris gradient SDS gel. Protein loading was checked with total protein stain for polyvinylidene fluoride (PVDF) membrane (Pierce MemCode reversible protein stain; Thermo Scientific, Waltham, MA, USA). Total cell lysate was prepared by addition of RIPA lysis buffer (Sigma Aldrich, St. Louis, MO, USA) supplemented with 1× Halt protease and phosphatase inhibitor cocktail (Thermo Scientific, Waltham, MA, USA). Cells were then washed three times with cold 1× PBS, incubated for 30 min on ice with RIPA buffer and centrifuged at 10,000 rpm for 5 min at 4 °C. Supernatants obtained were used for Western blot analysis. The protein content of supernatants was analyzed by utilizing a bio rad protein assay dye reagent based on the Bradford dye binding method (Catalog No. 500-0006, Bio-Rad, USA). SDS-PAGE with subsequent Western blot analysis was performed using the NuPage blotting kit (Life Technologies, Carlsbad, CA, USA). Ten µg of each sample (cell lysates) was loaded on a 4–12% Bis–Tris gradient gel and run at 200 V for 30 min. The proteins were then transferred to a PVDF membrane (Millipore, Billerca, MA, USA) and blocked with 5% non-fat skimmed milk in 1× phosphate buffered saline (PBS) with 0.05% tween (Sigma Aldrich, St. Louis, MO, USA). Antibodies against LL-37 were purchased from Innovagen, Sweden (polyclonal Rabbit, Cat No. PA-LL37-100) and Glyceraldehyde 3-phosphate dehydrogenase (GAPDH) from Santa Cruz Biotechnology, USA (polyclonal Rabbit, Cat No. sc-25778). The primary antibodies were diluted 1:1,000 in 1% non-fat skimmed milk powder (Blotto; Santa Cruz Biotechnologies, Santa Cruz, California, USA) in 0.05% Tween 1× PBS and incubated overnight at 4 °C. Next the membrane was incubated with 1:10.000 Horseradish Peroxidase (HRP)-linked secondary anti-rabbit IgG antibody (Cat No. A0545; Sigma Aldrich, St. Louis, MO, USA) in 0.05% Tween 1× PBS. The protein bands were visualized by chemiluminescence with Pierce ECLPlus Western blotting substrate (Thermo Scientific, Waltham, MA, USA) on Image Quant LAS 4000 station (GE Healthcare, Wauwatosa, WI, USA).

Immunofluorescence

VA10 cells were fixed with 3.5% paraformaldehyde (Sigma Aldrich, St. Louis, MO, USA) prepared in 1× PBS for 15 min. The cells were then washed twice with 1× PBS for 10 min at room temperature and blocked in immunofluorescence (IF) buffer (10% Fetal bovine serum (FBS) and 0.3% triton X-100 in 1× PBS) for 30 min. The cells were incubated overnight with LL-37 primary antibody (polyclonal Rabbit, Cat No. PA-LL37-100; Innovagen, Sweden) diluted 1:100 in the IF buffer, at 4 °C. Following overnight incubation, the cells were washed twice with 1× PBS. Next, the cells were incubated with secondary antibody anti-rabbit IgG Alexa Fluor® 488 (Catalog No. A11070)/546 conjugate (Catalog No. A11010) from Life Technologies, Carlsbad, CA, USA, 1:1,000 diluted in the IF buffer, for 1 h at room temperature. The cells were counterstained with nuclear stain 4, 6-diamidino-2-phenylindole (DAPI) at 1:5,000 dilution or 3 µM in IF buffer (Catalog No. D9564; Sigma Aldrich, St. Louis, USA). Finally, the cells were washed twice with 1× PBS and ultrapure H2O, and mounted in Fluormount-G solution (Southern Biotech, Birmingham, AL, USA) for microscopic analysis. The images were captured on Olympus fluoview Fv1200 confocal microscope at a 20× magnification. Olympus fluoview (FV) 1,000 software was used for processing the acquired images.

Oxidative stress measurement

After subjecting VA10 cells to cyclic stretch, CellROX green reagent (Life Technologies, Carlsbad, CA, USA) was added to the medium at a final concentration of 5 µM for 30 min. Next, the cells were washed twice with cold 1× PBS (1,000 rpm for 5 min) and detached with a 1× Accutase solution (Millipore, Billerca, MA, USA). Cells were then harvested and suspended in 100 µl of MACS buffer. (Miltenyi Biotec, San Diego, CA, USA) as per manufacturer’s instructions. The samples were analyzed in MACSQuant flow cytometer (Miltenyi Biotec, San Diego, CA, USA), placing the CellROX green reagent signal in FL1. Intact cells were gated in the Forward Scatter/Side Scatter plot to exclude debris. The resulting FL1 data was plotted on a histogram and is represented as % CellROX positive cells before and after cyclic stretch.

ELISA

Sandwich enzyme-linked immunosorbent assays (ELISAs) were performed utilizing an interleukin 8 (IL-8) and interferon gamma-induced protein 10 (IP-10) assay kit according to manufacturer’s instructions (Peprotech, London, UK). The results are represented from three independent experiments.

Statistical analysis

The q-RT PCR and ELISA results are represented as means ± standard errors of the means (S: E.) from three independent experiments. An unpaired Student’s t-test was used to compare two samples. P < 0.05 was considered statistically significant. All the statistical analysis for q-RT PCR and ELISA experiments was performed with the Prism 6 software (Graph Pad, USA). The Western blot and immunofluorescence data are represented from at least three independent experiments showing similar results.

Results

Cyclic stretch down-regulates the expression of the cathelicidin antimicrobial peptide

We screened for the effect of mechanical stretch on AMP expression. VA10 cells were subjected to stretch for 6 and 24 h to analyze early and late changes in AMP mRNA expression. The mRNA expression of AMPs cathelicidin (CAMP), human beta defensin-1 (DEFB1), Lactoferrin (LTF) and Lysozyme (LYZ) was analyzed with quantitative real time PCR (qRT-PCR). The basal mRNA expression of CAMP was decreased at both 6 h and 24 h after cell stretching (Fig. 1A). DEFB1 mRNA expression was reduced at 24 h after cell stretching but was unaffected after 6 h (Fig. S1). The basal mRNA expression of LTF and LZY was very low (Ct > 32) in the VA10 cells and was excluded from this study. The decrease in cathelicidin gene expression was further confirmed at protein level with Western blot (Fig. 1B) and immunofluorescence analysis (Fig. 1C). Western blot analysis of stretched VA10 cells showed a decrease in secreted pro-LL-37 (encoded by the CAMP gene) levels after 24 h of cyclic stretch (Fig. 1B). Further, immunofluorescence staining of stretched VA10 cells also showed a decrease in LL-37 protein expression at both 6 h and 24 h after stretching (Fig. 1C).

Figure 1 Cyclic mechanical stretch down-regulates cathelicidin antimicrobial peptide expression.

(A) VA10 cells were stretched for 6 and 24 h. The mRNA expression of cathelicidin antimicrobial peptide (CAMP) was analyzed with q-RT PCR after cell stretching (n = 3, mean ± S.E.). Relative expression levels (y-axis) in static cells were defined with an arbitrary value of ‘1’ and changes relative to this value in stretched samples are represented. (B) VA10 cells were subjected to cyclic stretch for 24 h. Cultured supernatants from stretched cells were used for analysis of secreted cathelicidin (pro-LL-37) protein expression by Western blot. Total protein loading is shown by staining with MemCode blue protein stain. The Western blot is a representative of three independent experiments showing similar results. (C) VA10 cells were stretched for 6 h and 24 h. The cells were then stained with antibody against LL-37 (green) and protein expression was visualized with immunofluorescence confocal microscopy. The cells were counterstained with nuclear stain DAPI (blue). Data is representative of three independent experiments showing similar results. Bar = 40 µm (ns indicates non-significant; p < 0.01, ∗∗; p < 0.001, ∗∗∗).

Treatment with vitamin D3 and/or 4-phenyl butyric acid (PBA) counteracts stretch mediated down-regulation of cathelicidin expression

We and others have demonstrated that treatment with vitamin D3 (1,25D3) and PBA enhances cathelicidin expression (Gombart, Borregaard & Koeffler, 2005; Kulkarni et al., 2015a; Kulkarni et al., 2015b). VA10 cells were treated with 100 nM 1,25D3 (Fig. 2A), 2 mM PBA (Fig. 2B) or co-treated with 2 mM PBA and 100 nM 1,25D3 (Fig. 2C). These cells were then stretched for either 6 h or 24 h and gene expression of CAMP was analyzed with q-RT PCR. Treatment of VA10 cells with 1,25D3 and/or PBA before stretch counteracted stretch mediated down-regulation of CAMP mRNA expression (Figs. 2A–2C). This counteraction was further confirmed at protein level with immunofluorescence (Fig. 2E) and Western blot analysis (Fig. 2F). Immunofluorescence staining of LL-37 confirmed that the 1,25D3 treatment prevented stretch mediated decrease in LL-37 protein expression with both at 6 h and 24 h stretch (Fig. 2E). Further, VA10 cells were treated with PBA and 1,25D3 as described above and stretched for 24 h. Protein expression of pro-LL-37 was analyzed with Western blot. Co-treatment with PBA and 1,25D3 predominantly enhanced pro-LL-37 expression in VA10 cells (Fig. 2F). This enhanced expression was lower in the stretched cells (Fig. 2F). Finally, we verified stretch mediated decrease of CAMP mRNA expression in another human bronchial epithelial cell line BCi. Similar to the VA10 cells, the BCi cells were treated with 100 nM 1,25D3 and stretched for 24 h. The basal mRNA expression of CAMP was reduced with 24 stretch, however was not significantly changed after 6 h. Similar to the results in VA10 cells, treatment with 1,25D3 counteracted stretch mediated down-regulation of CAMP mRNA expression in BCi cells (Fig. 2D). Thus, we demonstrate that treatment with 1,25D3 and PBA counteracted cyclic stretch mediated down-regulation of cathelicidin AMP expression.

Figure 2 Treatment with vitamin D3 (1,25D3) and 4-phenyl butyric acid (PBA) counteracts stretch mediated down-regulation of cathelicidin expression.

(A) VA10 cells were stretched for 6 and 24 h with (+) or without (−) 100 nM 1,25D3, (B) 2 mM PBA and (C) co-treated with vitamin D3 and PBA as shown in the figure. The mRNA expression of CAMP was assessed with qRT-PCR (n = 3, mean ± S.E.). Relative expression levels (y-axis) in static cells were defined with an arbitrary value of ‘1’ and changes relative to this value in stretched/treated samples are represented. (D) Similarly, the BCi cells were stretched for 6 h and 24 h with (+)/without (−) 100 nM 1,25D3 and the mRNA expression of CAMP was analyzed with q-RT PCR (n = 3, mean ± S.E.). Relative expression levels (y-axis) in static cells were defined with an arbitrary value of ‘1’ and changes relative to this value in stretched/treated samples are represented. (E) VA10 cells were treated with 20 nM 1,25D3 and stretched for 6 h and 24 h. LL-37 protein expression (red) was analyzed with immunofluorescence confocal microscopy. The cells were counterstained with nuclear stain DAPI (blue). The data is a representative of three independent experiments showing similar results. Bar = 100 µm. (F) Protein expression of cellular pro-LL-37 from stretched cells was also analyzed by Western blot analysis. VA10 cells were treated with 2 mM PBA, 20 nM 1,25D3 or co-treated with PBA and 1,25D3, followed by stretching for 24 h. GAPDH was used as a loading control. The Western blot is a representative of three independent experiments showing similar results (ns indicates non-significant; p < 0.05, ∗; p < 0.01, ∗∗; p < 0.001, ∗∗∗; p < 0.0001=∗∗∗∗).

Cyclic stretch activates a pro-inflammatory response and modulates toll-like receptor expression

Previous studies have demonstrated that VILI and in vitro cyclic stretching of alveolar lung epithelial cells activates a pro-inflammatory response by enhanced secretion of pro-inflammatory cytokines and chemokines (Vlahakis et al., 1999; Halbertsma et al., 2005). We studied the effect of cyclic stretch on inflammation in VA10 cells by screening for stretch mediated changes in pro-inflammatory cytokines and chemokines expression (Figs. 3A–3F). The VA10 cells were stretched for 6 h and 24 h. The mRNA expression of pro-inflammatory cytokines CXCL8 (encoding interleukin 8 or IL-8), IL1B (encoding interleukin 1 beta or IL-1β) and chemokines CXCL10 (encoding interferon gamma induced protein 10 or IP-10), CCL5 (encoding regulated on activation, normal T cell expressed and secreted or RANTES) was analyzed with q-RT PCR (Figs. 3A–3D) after stretching. The mRNA expression of genes encoding IL-8 and IL-1β was enhanced after 6 h and 24 h cyclic stretch (Figs. 3A–3C). Interestingly, mRNA expression of the gene encoding IP-10 was reduced following 6 h and 24 h stretch (Figs. 3A–3C). The mRNA expression of gene encoding RANTES was reduced after 24 h stretching and was not affected after 6 h stretch (Fig. 3D). Analysis of protein expression with ELISA demonstrated enhanced secretion of IL-8 after 24 h of cell stretching (Fig. 3E), whereas the total secreted levels of IP-10 was reduced (Fig. 3F). Increased oxidative stress through enhanced reactive oxygen species (ROS) has been shown to be involved in activation of a pro-inflammatory response (Mittal et al., 2014). We observed increased oxidative stress following cyclic stretch with the ROS detector dye Cell ROX. VA10 cells were stretched for 24 h and Cell ROX dye (5 µM) was added 30 min before end of cyclic stretching. The cells were then harvested and analyzed in a flow cytometer. A significant increase in percentage of Cell ROX positive cells was observed after stretching, indicating increased oxidative stress (Fig. 3G). Thus, we demonstrate that cyclic stretching activates a pro-inflammatory response in VA10 cells.

Figure 3 Cyclic stretch activates a pro-inflammatory response and enhances oxidative stress.

(A–D) VA10 cells were subjected to stretch for 6 and 24 h. The mRNA expression of genes encoding pro inflammatory cytokines IL-8 (A), IL-1β (B) and chemokines IP-10 (C), RANTES (D) was measured with q-RT PCR (n = 3, mean ± S.E.). Relative expression levels (y-axis) in static cells were defined with an arbitrary value of ‘1’ and changes relative to this value in stretched samples are represented. (E–F) The protein expression of IL-8 (E) and IP-10 (F) from cultured supernatants was measured with ELISA. VA10 cells were stretched for 24 h and ELISAs were performed (n = 3, mean ± S.E.). (G) Oxidative stress was measured with CellROX green reagent. VA10 cells were subjected to stretch for 24 h. CellROX dye (5 µM) was added 30 min before the end of stretching. The cells were then harvested and analyzed by flow cytometry. The data is represented as percentage positive CellROX (ROS) cells before and after cyclic stretch (n = 3, mean ± S.E.) (ns indicates non-significant; p < 0.05, ∗; p < 0.01, ∗∗; p < 0.001, ∗∗∗; p < 0.0001, ∗∗∗∗).

Next, we screened for stretch mediated changes in toll-like receptor (TLR) expression in VA10 cells (Figs. 4A–4H). TLRs play an important role in activation of pro-inflammatory responses and have been shown to be modulated by mechanical stretching of cells (Takeda & Akira, 2005; Shyu et al., 2010). VA10 cells were stretched for 6 h and 24 h as described above. The mRNA expression of TLR1, TLR2, TLR3, TLR4, TLR5, TLR6, TLR8 and TLR9 was analyzed with q-RT PCR. The mRNA expression of TLR3, TLR5 and TLR8 was reduced 6 h after stretch (Figs. 4C, 4E and 4G). Further, the mRNA expression of TLR2 (Fig. 4B) was increased and gene expression of TLR3 (Fig. 4C) and TLR9 (Fig. 4H) was decreased after 24 h stretch.

Figure 4 Cyclic stretch modulates toll-like receptor (TLR) gene expression.

(A–H) VA10 cells were subjected to cyclic stretch for 6 h and 24 h. The mRNA expression of TLR1 (A), TLR2 (B), TLR3 (C), TLR4 (D), TLR5 (E), TLR6 (F), TLR8 (G) and TLR9 (H) was analysed with q-RT PCR (n = 3, mean ± S.E.). Relative expression levels (y-axis) in static cells were defined with an arbitrary value of ‘1’ and changes relative to this value in stretched samples are represented (ns indicates non-significant; p < 0.05, ∗; p < 0.01, ∗∗; p < 0.0001, ∗∗∗∗).

Treatment with vitamin D3 and PBA differentially affects stretch mediated changes in pro-inflammatory cytokine IL-8 expression

Vitamin D3 and PBA have been shown to differentially modulate inflammatory responses, mainly having anti-inflammatory effect (Hansdottir et al., 2010; Roy et al., 2012). VA10 cells were treated with 20 nM 1,25D3 (Fig. 5A) and 2 mM PBA (Fig. 5B), followed by stretching for 6 h and 24 h as shown in Fig. 5. The mRNA expression of genes encoding the pro-inflammatory cytokine IL-8 was analyzed with q-RT PCR. Treatment with 1,25D3 and PBA increased basal expression of IL-8 after 6 h treatment in static cells (Figs. 5A and 5B). Interestingly, treatment with PBA significantly enhanced stretch mediated increase in IL-8 gene expression after 6 h stretch (Fig. 5B). This enhancement was not observed at 24 h after stretch (Fig. 5B).

Figure 5 Treatment with 1,25D3 and PBA differentially affects stretch mediated changes in pro-inflammatory cytokine IL-8 gene expression.

(A, B) VA10 cells were treated with (+) or without (−) 20 nM 1,25D3 (A) or 2 mM PBA (B) and subjected to cyclic stretch for 6 h and 24 h. The mRNA expression of genes encoding pro-inflammatory cytokine IL-8 was analyzed with q-RT PCR (n = 3, mean ± S.E.). Relative expression levels (y-axis) in static cells were defined with an arbitrary value of ‘1’ and changes relative to this value in stretched/treated samples are represented. (ns indicates non-significant; p < 0.05, ∗; p < 0.01, ∗∗; p < 0.0001, ∗∗∗∗).

Discussion

Mechanical ventilation (MV) is necessary for maintenance of gas exchange and to prevent cardiorespiratory collapse in patients suffering from serious respiratory failure and the more severe condition of acute respiratory distress syndrome (ARDS), but may also result in increased mortality of patients by ventilator induced lung injury (VILI) (Slutsky & Ranieri, 2013). Cyclic stretch generated during MV has been implicated in modulation of innate immune responses leading to side effects that include biotrauma and secondary infections like pneumonia (Santos et al., 2005). We hypothesized that cyclic stretch modulates the expression of antimicrobial peptides (AMPs) that constitute and important arm of the innate immune system. MV can be modelled in vitro by cyclic mechanical stretching of respiratory epithelial cells (Vlahakis et al., 1999; Pugin et al., 2008). In this study we show that cyclic mechanical stretch of human bronchial respiratory epithelial cell line VA10 down-regulates gene expression of AMP cathelicidin and human beta defensin 1 (Figs. 1 and S1). To our knowledge, this is first report demonstrating direct effects of mechanical stretch on AMP expression in vitro. AMPs prevent the invasion of pathogens through the lung epithelium and an impaired AMP production could render the tissues more susceptible to infections (Bals et al., 1998). In immunocompromised mice, CRAMP (homologue of human cathelicidin) knockout resulted in increased susceptibility to dermal and respiratory infections (Nizet et al., 2001; Kovach et al., 2012). Our group has previously demonstrated that the diarrhoeal pathogen Shigella down-regulates cathelicidin antimicrobial peptide (CAMP) expression as a strategy to subvert innate immune responses (Islam et al., 2001). This down-regulation can be counteracted by treatment with inducers of cathelicidin like butyric acid and PBA (Raqib et al., 2006; Sarker et al., 2011). In this study, we demonstrate that treatment with vitamin D3 (1,25D3) and PBA counteracts cyclic stretch mediated down-regulation of cathelicidin expression in the VA10 cells (Fig. 2). Interestingly, patients with low vitamin D3 levels admitted to intensive care units receiving MV had higher mortality rate and longer hospital stay than patients with sufficient vitamin D3 (Parekh, Thickett & Turner, 2013; Leaf et al., 2015; Quraishi et al., 2015). Further, in animal models, MV has been shown to promote bacterial dissemination and infections (Schortgen et al., 2004). Thus induction of AMPs with inducers like 1,25D3 and PBA may be a useful strategy to reduce these complications arising from MV, preventing infections and hence can be sought as adjunct therapeutics.

We further studied the effects of cyclic stretch on inflammatory responses in the VA10 cells. Lung injury in MV patients is aggravated due to activation of MV mediated sterile inflammatory response and may lead to biotrauma (Santos et al., 2005; Dhanireddy et al., 2006). This damage due to MV activated inflammatory responses is further enhanced during infections. A recent meeting abstract noted that rhinovirus infection enhanced cyclic stretch mediated up-regulation of pro-inflammatory cytokine expression in human bronchial epithelial cells (Nikitenko et al., 2014). We noticed a significant increase in mRNA expression of genes encoding the pro-inflammatory cytokines IL-8 and IL-1β in the VA10 cells subjected to cyclic stretch (Fig. 3). Interestingly, the gene expression of chemokine IP-10 reduced at both 6 and 24 h of stretch. Further the mRNA expression of chemokine RANTES was down-regulated after 24 h of stretch (Fig. 3). Others have similarly shown cyclic stretch mediated up-regulation of IL-8 and down-regulation of RANTES expression in BEAS-2B human bronchial epithelial cell line (Oudin & Pugin, 2002; Thomas et al., 2006). Stretch mediated increase in IL-8 levels was shown to be dependent on activation of mitogen activated protein kinase (MAPK) and rho-kinase signaling (Oudin & Pugin, 2002; Thomas et al., 2006). Further, secretion of IL-1β was shown to be enhanced in mouse alveolar macrophages following MV induced cyclic stretch via caspase-1 and TLR 4 dependent activation of NLRP3 inflammasomes (Wu et al., 2013).

Increased oxidative stress following cyclic stretch further confirmed activation of a pro-inflammatory response in the VA10 cells (Fig. 3). Recently, cyclic stretch was shown to activate mitochondrial reactive oxygen species (ROS) production via activation of nuclear transcription factor NFκB and NADPH oxidase (Nox) 4 signaling in pulmonary arterial smooth muscle cells (Wedgwood et al., 2015). Mitochondrial ROS drives production of pro-inflammatory cytokines (Naik & Dixit, 2011). We stained VA10 cells with the CellROX (ROS detector) dye and noticed a significant increase in percentage of CellROX positive VA10 cells after cyclic stretch, indicating enhancement of oxidative stress (Fig. 3). We further looked at the effects of cyclic stretch on expression of toll-like receptors (TLRs). TLRs sense pathogen associated molecular patterns (PAMPs) upon infection, leading to activation of down-stream signaling pathways and induction of pro-inflammatory cytokines and chemokines (Takeda & Akira, 2005). Cyclic stretch of cultured cardiomyocytes was shown to enhance TLR4 gene expression via activation of the p38 MAPK and NF-κB pathway (Shyu et al., 2010). Interestingly, cyclic stretch of human alveolar epithelial cell line A549 enhanced TLR2 expression (Charles et al., 2011). Further it was demonstrated that cyclic stretch enhanced IL-6 and IL-8 secretion in response to Pam3CSK4, a classical TLR2 ligand (Charles et al., 2011). In our study, the mRNA expression of TLR3, TLR5, TLR8 and TLR9 was down-regulated after cyclic stretch, whereas the gene expression of TLR2 was increased in the VA10 cells (Fig. 4). A direct causal link between the stretch mediated changes in TLR gene expression and pro-inflammatory cytokine expression needs to be established. Vitamin D3 and PBA have been shown to have anti-inflammatory effects (Hansdottir et al., 2010; Roy et al., 2012). In our study, treatment of the VA10 cells with both 1,25D3 and PBA enhanced mRNA expression of the gene encoding IL-8 (Fig. 5). This is in correlation with our previous study in the VA10 cells (Kulkarni et al., 2015a; Kulkarni et al., 2015b). Further, treatment with PBA significantly enhanced stretch mediated increase in IL-8 gene expression 6 h after stretching in the VA10 cells (Fig. 5).

Our study has certain limitations. (1) The study was performed exclusively in cell lines. However, these respiratory cell lines (VA10 and BCi) have been shown to have primary cell like characteristics and have differentiation potential when cultured at an air–liquid interface (Halldorsson et al., 2007; Walters et al., 2013). They represent the upper airway lung epithelia. Primary human bronchial epithelial cells did not grow properly on collagen I coated bioflex silastic membranes and had to be excluded from this study. (2) The mechanism behind cyclic stretch mediated down-regulation of AMP expression needs to be elucidated and is a future area of interest. We hypothesize that stretch activated stress pathways (e.g., hypoxia related HIF-1α (Eckle et al., 2013; Fan et al., 2015)) could be involved in the observed down-regulation of AMP expression. Interestingly, acidification of cellular milieu upon cyclic stretch has been shown to promote bacterial growth in lung epithelial cells (Pugin et al., 2008). The relationship between stretch altered pH and its effects on AMP gene expression is also an area of interest.

In conclusion, our in vitro data shows that cyclic stretch down-regulates the expression of AMP cathelicidin in VA10 and BCi respiratory epithelial cells and activates a pro-inflammatory response in VA10 cells. These results could have clinical implications in regards to ventilator treatment of patients by identifying ways to increase the endurance of lung tissues to mechanical strain and preventing respiratory infections, encouraging further in vivo studies in this field.

Supplemental Information

Figure S1 Cyclic stretch down-regulates human beta defensin 1 (DEFB1) gene expression

VA10 cells were subjected to cyclic stretch for 6 h and 24 h. The mRNA expression of DEFB1 was analyzed with q-RT PCR (n = 3, mean ± S.E. ). Relative expression levels (y-axis) in static cells were defined with an arbitrary value of ‘1’ and changes relative to this value in stretched samples are represented. (ns indicates non-significant; p < 0.01 = ∗∗).

Click here for additional data file.

Supplemental Information 2 Raw data

Click here for additional data file.

We would like to thank Jon Thor Bergthorsson and Katrin Birna Petursdottir for the help with analyzing flow cytometry data. We would like to specially thank Arí Jon Arason for introduction to the Flexcell tension system.

Additional Information and Declarations

Competing Interests

Author Contributions

Data Availability

The authors declare there are no competing interests.

Harpa Karadottir and Nikhil Nitin Kulkarni conceived and designed the experiments, performed the experiments, analyzed the data, wrote the paper, prepared figures and/or tables.

Thorarinn Gudjonsson and Sigurbergur Karason analyzed the data, contributed reagents/materials/analysis tools, reviewed drafts of the paper.

Gudmundur Hrafn Gudmundsson conceived and designed the experiments, analyzed the data, contributed reagents/materials/analysis tools, wrote the paper.

The following information was supplied regarding data availability:

The research in this article did not generate any raw data.

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
