# Peer review of "Cyclic mechanical stretch down-regulates cathelicidin antimicrobial peptide expression and activates a pro-inflammatory response in human bronchial epithelial cells"

_PeerJ, doi:10.7717/peerj.1483_

## Round 0.1 · original submission · Major Revisions

This manuscript holds merits; however, there are a few concerns that must be addressed before acceptance for publication in PeerJ. I suggest that a second cell line or other model systems may be included to verify the current findings.

Reviewer 1 ·

Basic reporting

Karadottir et al. show basically in VA10 cells that a cyclic stretch of the cells results in a downregulation of cathelicidin antimicrobial peptide (CAMP) gene expression and pro-LL-37 protein expression after 6 and 24 hours. Furthermore, they show that a treatment with vitamin D3 (1,25D3) and 4-phenyl butyric acid (PBA) and a combination of both stimuli counteracts the stretch mediated down-regulation of cathelicidin expression in VA10 cells on RNA and protein level. For 1,25D3 they confirm it on RNA level using the cell line BCi NS1.1. The gene expression of the pro-inflammatory cytokines IL-8 and IL-1B was enhanced whereas the gene expression of the chemokines IP-10 and RANTES was downregulated. On protein level, secreted IL-8 was enhanced whereas IP-10 was decreased and oxidative stress levels were elevated in VA10 cells. Furthermore, toll-like receptor (TLR) gene expression of TLR2 was enhanced after 24 hours cyclic stretch, and TLR3 and TLR9 gene expression was reduced after 24 hours. TLR8 gene expression was reduced after 6 hours. Finally, Karadottir et al. show that the treatment of stretched VA10 cells with 1,25D3 and PBA increases the IL-8 mRNA expression.
Introduction: lines 89-92: one more recent publication should be included: Kulkarni et al., 2015 “Glucocorticoid dexamethasone down-regulates basal and vitamin D3 induced cathelicidin expression in human monocytes and bronchial epithelial cell line.” Immunobiology, doi: 10.1016/j.imbio.2015.09.001 as it was already shown there that in BCi NS 1.1 cells, Vitamin D3 upregulates CAMP expression.
Figures: in Figures 1, 2, 3, 4, 5 all real-time PCR-based figures need an appropriate y-axis labelling, relative expression is not sufficient as it can mean anything; it is common to give a log2-based scale
For all figures showing error bars: authors should mention in each figure legend that they were using mean +/- standard error
Figure 3: * p<0.05 is missing in figure legend
Figure 4: ***p<0.001 is not shown in any graph, so remove it from the figure legend
Figure 5: ***p<0.001 is not shown in any graph, so remove it from the figure legend

Experimental design

The authors state that cyclic stretch downregulated CAMP gene expression in VA10 and BCi NS 1.1 cells (Abstract line 35-36; introduction lines 97-98). In discussion part, they claim that AMPs are downregulated in epithelial cells and pro-inflammatory response can after cyclic stretch (discussion, line 386-387). But how can such statements be given if obviously only VA10 cells were analyzed?
At least some major findings like downregulation of CAMP gene expression after cyclic stretch as well as not only after Vitamin D3 treatment (Figure 2D), but also after PBA treatment and after the combination of Vitamin D3 and PBA treatment should be shown to made data more solid.
And they should give a statement/explanation why they focused on VA10 cells and not on BCi NS1.1 cells in the further study.
Lines 163-168: For transparence, order numbers of all antibodies should be given by the authors.
And the SDS-PAGE and Western Blot paragraph should be looked on as 150-151 is redundant to line 160-161 as there is no difference in loading supernatants or sample, so this can be summarized into 1 statement.
Line 158: with protein assay reagent was taken for analyzing protein content?
Line 181: what was the final concentration of DAPI? A 1:5000 dilution can mean anything and depends on the initial concentration which is not given either.

Validity of the findings

No comments.

Additional comments

lines 430-433: year is missing for this reference
lines 453-454: formatting has to be checked as “2^-delta delta CT method” is not written properly.

Reviewer 2 ·

Basic reporting

The manuscript entitled cyclic mechanical stretch down-regulates cathelicidin antimicrobial peptide expression and activates a pro-inflammatory response in human bronchial epithelial cells” represents a well planned study on a potentially important aspect of patient care and clinical intervention. The results presented in this manuscript may be of interest to a wide base of readers with research interest in patient care, anti-microbial peptides to nasocomial infections etc.

Authors have demonstrated how cyclic stretch (which mimics the effect of mechanical ventilation) modulates the expression of an Anti-microbial peptide (Cathelicidin). Further authors have also demonstrated that the applied cyclic stretch activates expression of pro-inflammatory cytokines, enhances oxidative stress and modulates gene expression of toll- like receptors (TLRs).

The other major finding reported in this manuscript is protective effect of Vitamin D3 and PBA for amelioration of above adverse effects of mechanical stretch. This finding suggests that Vitamin D3, PBA and/ or their derivatives could be developed as potential treatment for patients on Mechanical Ventilation.

Overall, the manuscript has been prepared thoroughly and reads quite well for scientific content as well as presentation.

Experimental design

The experimental design used for the study is appropriate and technically sound. Authors have used in vitro assay model system (Bronchial epithelial cell lines) to test the effect of cyclic stretch on different functions e.g. AMP expression, immune response modulation, TLR expression etc.,

The findings have been verified appropriately with use of complementary experimental approaches (e.g. qRT-PCR, Western Blotting, ELISA and immunofluorescence analyses).

Also, authors have take adequate care for experimental replicates and the data has been presented after performing the necessary statistical analyses.

Validity of the findings

The findings reported in this manuscript have been validated by complementary experimental approaches. Most of the findings reported in this manuscript are in concurrence with previous reports in this field of research.

Additional comments

Major concern:

1: Abstract: Authors claim that ‘down- regulation of cathelicidin expression was coupled with an increase in pro-inflammatory response’. This statement may be toned down a bit since it is not evident from results whether these observations are mechanistically coupled with one another or they are concurrent – independent responses of cyclic stretch.

2: Introduction: The Introduction section should be extended to briefly mention the potential effect of mechanical ventilation on immune response modulation, activation of oxidative stress and effect on cell signaling pathways. In its current state, the introduction section does not clearly justify as to why these parameters have been evaluated in the manuscript.

3: Materials and Methods: This section is quite thorough and elaborative

4: Results: Results are presented adequately and don’t need any significant correction, Yet it would useful if authors consider the following:

(i): Figure 1B: the hybridization signal for internal control (GAPDH) should be shown.

(ii) Figure 2: while results from qRT-PCR and immunofluorescence indicate that treatment of 1,25D3 and PBA counteract the cyclic stretch mediated down- regulation of cathelicidin expression, the western blot analyses does not reflect the same.

(iii) Figure 3, 4 & 5: It would be interesting to investigate if treatment of 1,25D3 and PBA also induces changes in IP-10, RANTES, ROS% cells and TLR expression levels.

5: Discussion: Overall, the discussion is rather lengthy and reads somewhat repetitive. It could be shortened to focus on discussion of mechanism rather than repeating the results. For examples authors may discuss whether the different functions investigated during this study are mechanistically related to one another.

Suggested Experiment: The following experiment may be beyond the scope of the present study; however, it would be extremely useful and conclusive if authors can perform a ‘comparative transcriptome analyses’ of cyclic stretch v/s cyclic stretch + 1,25D3 and/or PBA cells for human cytokine specific and AMP specific transcriptome.

·

Basic reporting

An overall excellent study, well written.

Experimental design

Well done

Validity of the findings

Totally agree with their conclusions

Additional comments

The fascinating study by Karadottir et al explores the effects of mechanical stretch on human bronchial epithelial cells in an in vitro system. The authors show that repeated cyclic stretching leads to a decrease in the expression of cathelicidin, an increase in the expression of pro-inflammatory cytokines, an increase in oxidative stress, and a change in the pattern of expression of TLRs. Exposure of the cells to either Vit D or PBA prevented the drop in LL-37 expression in response to stretching. The effects of these agents on IL-8 expression was more complex, exhibiting a suppressive effect after longer periods of stretch.
This is an absolutely fascinating study. It demands a clinical trial of the administration of either VitD and or PBA in survival and morbidity of ventilated patients.
The ms is well written, the Methods very clear, and the References ok. The Figures are well presented.
I had one minor question. In the absence of the addition of Vit D or PBA the stretched bronchial epithelial cells seem to lose their immunofluorescence (Figure 1C). What happens to the peptide? Is it degraded, or secreted into the medium. Does stretching naturally cause a release of peptide, as might be expected in the setting of a normal respiratory cycle? Could the authors please clarify this.
This is a wonderful story with profound medical implications.

---

## Round 0.2 · Minor Revisions

I am pleased to inform you that the manuscript is suitable for publication in PeerJ. Before accepting the manuscript officially, please make the final changes suggested by reviewer 1.

Reviewer 1 ·

Basic reporting

The manuscript demonstrates that a mechanical stretch down-regulates cathelicidin antimicrobial peptide expression and activates a pro-inflammatory response in the human bronchial epithelial cell line model VA10.

Thanks for critically correcting the manuscript and adding the desired information.

Experimental design

no comments.

Validity of the findings

no comments.

Additional comments

Just a very few comments:

Lines 197 & 198: it has to be “Miltenyi”, the 1 afterwards should be removed

Figures 1-5: the analysis and data interpretation using delta 2^-delta delta ct method is absolutely fine, it is a valid and frequently used method, there is no doubt at all! But still you have to mention this on the axis as a figure should be self-understandable for the readership even without the main text. Therefore for axis labelling it should be mentioned that it is a logarithmic calculation with basis 2 as e.g. a fold change of 4 means in the end 2^4 = 16 fold difference in the RNA amount and not 4 fold difference as you could assume in a decimal system.

Reviewer 2 ·

Basic reporting

The revised version of manuscript is fine and makes a good scientific reading.

Experimental design

Well planned and executed experiments. The results obtained and reported in the manuscript are suitable for conclusions drawn.

Validity of the findings

Findings of the present study have been duly validated with complementary experiments.

Additional comments

The overall presentation and scientific content of the manuscript is quite good. The revised version reads much more comprehensive and therefore, it is quite suitable for publication.

·

Basic reporting

I am happy with the revised version

Experimental design

Excellent

Validity of the findings

Important

Additional comments

An important observation with great potential medical value

---

## Round 0.3 · accepted · Accept

I am pleased to inform that the manuscript is suitable for publication in PeerJ.

---

## Author Rebuttal · Round 0.3

17.11.2015

Reykjavik, Iceland.

To

Dr. Pankaj Goyal,

Editor PeerJ,

Dear Editor,

We would like to thank all the reviewers for their positive comments on our manuscript. We have attached a response to the comments from reviewer 1. Although, reviewer 1 suggests representing the Y-axis for q-RT PCR data on a $\log_2$ scale, we would like to draw your attention to the fact that the data as presented now on Y-axis is not log transformed. Hence we cannot label Y-axis as $\log_2$ relative expression. We have explained in details the interpretation of Y-axis in figure legends for the readers to have more clarity. The method we used for representing q-RT PCR data is widely accepted, just like the $\log_2$ based representation.

All the authors have read and approved the final version of this manuscript.

Sincerely,

Gudmundur Hrafn Gudmundsson, PhD
Professor of Cell Biology,
Biomedical Center,
University of Iceland,
101 Reykjavik,
Iceland.

Thank you for your submission to PeerJ. I am writing to inform you that in my opinion as the Academic Editor for your article, your manuscript "Cyclic mechanical stretch down-regulates cathelicidin antimicrobial peptide expression and activates a pro-inflammatory response in human bronchial epithelial cells." (#2015:09:6825:1:0:REVIEW) requires some minor revisions before we could accept it for publication.

The comments supplied by the reviewers on this revision are pasted below. My comments are as follows:

## Editor's comments

I am pleased to inform you that the manuscript is suitable for publication in PeerJ. Before accepting the manuscript officially, please make the final changes suggested by reviewer 1.

**Response: We would like to thank the reviewers for their positive reviews on our manuscript. We have discussed with all the co-authors regarding the representation of the q- RT PCR results on a log scale on the Y-axis. Although, the data can be represented on a $\log_2$ scale as suggested by reviewer 1, representing the data as 'relative expression' on Y-axis is also a standard and well accepted method. Please note these published articles where the q-RT PCR data is represented with the term 'relative expression', 'fold difference' or 'fold change' [ 1) PMID: PMID: 20199660, Figure 2B and 2C); 2) PMID: 19939273, Figure 2A and 2B; and an article published in PeerJ where the Y-axis is labeled as Fold Expression, PMID: 23646287, figure 1A-1F)]. We have also added an extra sentence in the figure legends for all the q-RT PCR results, so that the readers are not confused about the method of representation ('Relative expression levels (y-axis) in static untreated cells were defined with an arbitrary value of '1' and changes relative to this value in stretched/treated samples are represented.'). We hope this clarifies the comments from reviewer 1. We have also made other changes suggested by reviewer 1.**

If you are willing to undertake these changes, please submit your revised manuscript (with any rebuttal information*) to the journal within 45 days.

Pankaj Goyal
Academic Editor for PeerJ

# Reviewer Comments

## Reviewer 1 (Anonymous)

## Basic reporting

The manuscript demonstrates that a mechanical stretch down-regulates cathelicidin antimicrobial peptide expression and activates a pro-inflammatory response in the human bronchial epithelial cell line model VA10.

Thanks for critically correcting the manuscript and adding the desired information.

## Experimental design

no comments.

## Validity of the findings

no comments.

## Comments for the author

Just a very few comments:

Lines 197 & 198: it has to be "Miltenyi", the 1 afterwards should be removed

**Response: We would like to thank the reviewer for pointing out this error. This has now been corrected.**

Figures 1-5: the analysis and data interpretation using delta 2^-delta delta ct method is absolutely fine, it is a valid and frequently used method, there is no doubt at all! But still you have to mention this on the axis as a figure should be self-understandable for the readership even without the main text. Therefore for axis labelling it should be mentioned that it is a logarithmic calculation with basis 2 as e.g. a fold change of 4 means in the end $2^4 = 16$ fold difference in the RNA amount and not 4 fold difference as you could assume in a decimal system.

**Response: The term 'relative expression' refers to the relative expression of target gene with respect to untreated control, where untreated control is represented with an arbitrary value of '1' and changes in treated samples are represented as changes relative to this untreated control (change relative to '1'). Representing the Y-axis with the term 'Relative expression' is a standard practice. In order to label the Y-axis as Fold change ($\log_2$) we will have to transform all our data to a log scale. This would not add any significant information to our results. Nevertheless, we are aware that q-RT PCR data can be**

represented on a log scale and in that case instead of the control untreated sample having the arbitrary value of '1', this sample will have a value of '0' and fold changes that are down-regulated will be represented on a negative scale (commonly referred to as log scale on Y-axis). Thus, we think the method we have used represents data in a standard manner where all the controls have a relative value of ´1' and changes relative to this value are represented. Please note these published articles where the q-RT PCR data is represented with the term 'relative expression', 'fold difference' or 'fold change' [ 1) PMID: PMID: 20199660, Figure 2B and 2C); 2) PMID: 19939273, Figure 2A and 2B; and an article published in PeerJ where the Y-axis is labeled as Fold Expression, PMID: 23646287, figure 1A-1F)]. We also used the same Y-axis labeling in our recent published article: PMID:26358366. In all the cases the data on the Y-axis is not log transformed. Thus one of the many ways to represent the q-RT PCR data is to simply label the Y-axis with 'Relative expression' and mention the target gene. This is widely accepted by readers familiar with q-RT PCR data. We have now added in the figure legends 'Relative expression levels (y-axis) in static untreated cells were defined with an arbitrary value of '1' and changes relative to this value in stretched/treated samples are represented.'. We hope this explanation will be sufficient for the reader for clarity.

## Reviewer 2 (Anonymous)

### Basic reporting

The revised version of manuscript is fine and makes a good scientific reading.

### Experimental design

Well planned and executed experiments. The results obtained and reported in the manuscript are suitable for conclusions drawn.

### Validity of the findings

Findings of the present study have been duly validated with complementary experiments.

### Comments for the author

The overall presentation and scientific content of the manuscript is quite good. The revised version reads much more comprehensive and therefore, it is quite suitable for publication.

**Response: We would like to thank the reviewer 2 for the positive comments on the manuscript.**

## Reviewer 3 (Michael Zasloff)

### Basic reporting

I am happy with the revised version

### Experimental design

Excellent

### Validity of the findings

Important

### Comments for the author

An important observation with great potential medical value

**Response: We would like to thank Dr. Michael Zasloff for the positive comments on the manuscript.**